# Outcomes of Salvage Treatment After Primary Treatment for Renal Cell Cancer: A Systematic Review

**DOI:** 10.3390/diagnostics15070838

**Published:** 2025-03-25

**Authors:** Nicola Longo, Francesco Di Bello, Luigi Napolitano, Ernesto Di Mauro, Simone Morra, Giuliano Granata, Federico Polverino, Agostino Fraia, Gabriele Pezone, Roberto La Rocca, Claudia Collà Ruvolo, Gianluigi Califano, Massimiliano Creta

**Affiliations:** Department of Neurosciences, Science of Reproduction and Odontostomatology, University of Naples Federico II, 80055 Naples, Italy; nicola.longo@unina.it (N.L.); dr.luiginapolitano@gmail.com (L.N.); ernestodm9@gmail.com (E.D.M.); simonemorra@outlook.com (S.M.); giuliano.granata97@gmail.com (G.G.); fedepolve97@gmail.com (F.P.); agostino.fraia@gmail.com (A.F.); gabriele.pezone@unina.it (G.P.); roberto.larocca@gmail.com (R.L.R.); c.collaruvolo@gmail.com (C.C.R.); gianl.califano2@gmail.com (G.C.); max.creta@gmail.com (M.C.)

**Keywords:** RCC, partial nephrectomy, Clavien–Dindo, complications

## Abstract

**Background/Objective**: The grade of recommendation for renal cell carcinoma (RCC) salvage treatment options is weak. The aim of the current study is to summarize available evidence about the surgical, oncological, and functional outcomes of salvage renal options after previous treatments for RCC. **Methods**: A systematic search (PROSPERO: CRD42024618629) was performed according to the PRISMA statement. A pooled analysis was performed to quantify the effect size (ES) for an overall postoperative Clavien–Dindo (CD) grade ≥ III and postoperative and intraoperative complications for either partial or radical nephrectomy (PN or RN) subgroups. **Results**: Overall, 11 studies involving 331 patients were included in the final analysis. The median age ranged from 44 to 75 years. Primary treatments for RCC included 267 (80.6%) PNs, 40 (12.0%) radiofrequency ablations (RFAs), 23 (6.9%) cryotherapies, and 1 (0.3%) stereotactic radiotherapy. Overall, the median time of local recurrence for RCC patients was from 25 to 99 months for PNs, 13 months for RFA and cryotherapy, and 6 months for stereotactic radiotherapy. The overall pooled analysis was associated with an ES of 0.28 for overall postoperative complications and of 0.11 for CD ≥ III postoperative complications (all *p* < 0.001). Within subgroup analyses, RN was invariably associated with less complications (the ES ranged from 0.05 to 0.17). **Conclusions**: The profile of oncological safety as well as the morbidity and mortality of salvage treatment options for RCC may justify considering such procedures in a salvage setting.

## 1. Introduction

Renal cell carcinoma (RCC) accounts for about 3% of all cancers and is the most common solid lesion within the kidney accounting for nearly 90% of all kidney malignant tumors [1]. Surgery represents the gold standard treatment for localized RCC with partial nephrectomy (PN) being strongly recommended in patients with T1 tumors and weakly suggested in patients with T2 tumors and a solitary kidney or chronic kidney disease, if technically feasible [1,2,3]. Alternative tumor ablation techniques including cryoablation and radiofrequency ablation are weakly recommended in frail and/or comorbid patients with small renal masses [1,4]. Taken together, nephron-sparing techniques aim to preserve renal function by eliminating localized disease while preserving healthy tissue [5,6,7]. However, local relapse may occur due to an incomplete resection of the primary tumor, multifocality, or the local spread of the tumor by microvascular embolization [1,3,5,8]. It has been reported that local cancer recurrence can occur in a percentage of patients undergoing PN that ranges between 1 and 10% for low-risk disease and up to 60% for high-risk disease at 5 years postoperatively [5,7,9,10]. Following radiofrequency ablation and cryoablation, the rate of local recurrences is approximately 8% and 5%, respectively [11,12,13]. With the increased utilization of PN and ablation, an increased need for the management of patients with ipsilateral tumor recurrence is expected [8,14,15]. The management of local cancer recurrence in patients who underwent previous nephron-sparing approaches represents a surgical challenge mainly due to the scarring, fibrosis, and subsequent obliteration of surgical planes [5,9]. Salvage procedures were firstly described in 2008 by Magera et al. [12]. Currently, salvage options after nephron-sparing techniques for RCC include partial or radical nephrectomy with either open or robotic surgical approach and non-surgical ablative techniques [16]. The European Association of Urology Guidelines recommend local treatment of locally recurrent disease when technically possible and after balancing adverse prognostic features, comorbidities, and life expectancy [1]. However, the level of evidence is still low, and the grade of recommendation is weak [1]. Moreover, the most valid salvage procedure has not yet been defined [1]. We aimed to summarize available evidence about the surgical, oncological, and functional outcomes of salvage renal options after previous treatments for RCC.

## 2. Materials and Methods

The present analysis was conducted and reported according to the general guidelines recommended by the Primary Reporting Items for Systematic Reviews and Meta-Analyses (PRISMA) statement (Appendix A) [17]. This protocol was registered in PROSPERO (CRD42024618629).

### 2.1. Literature Search

The search was performed on the Medline (US National Library of Medicine, Bethesda, MD, USA), Scopus (Elsevier, Amsterdam, The Netherlands), and Web of Science Core Collection (Thomson Reuters, Toronto, ON, Canada) databases up to May 1st. The following terms were combined to capture relevant publications: “renal cell carcinoma” OR “Renal carcinoma” AND “local recurrence” OR “Salvage radical nephrectomy” OR “Salvage partial nephrectomy” OR “Salvage thermal nephrectomy”. Reference lists in relevant articles and reviews were also screened for additional studies. Conference abstracts were also considered.

### 2.2. Selection Criteria

Two authors (F.D.B. and N.L.) reviewed the records separately and individually to select relevant publications, with any discrepancies resolved by a third author (M.C.). To assess eligibility for this systematic review, PICOS (participants, intervention, comparisons, outcomes, and study type) criteria were used to assess the eligibility of studies. PICOS criteria were set as follows: (P) patients with disease relapse previously underwent treatment for RCC; (I) salvage radical nephrectomy (RN), salvage nephron-sparing surgery (NSS), salvage radiotherapy (RT), salvage radiofrequency ablation (RFA), salvage cryoablation, salvage therapies; (C): none; (O): surgical, oncological, and functional parameters; (S) conference abstracts, case reports, case series, randomized clinical trials, retrospective, cross-sectional, and prospective studies.

### 2.3. Data Collection

The following data were extracted from eligible studies: first author, study design, study year, sample size, study quality, patients’ age, gender, body mass index (BMI), ethnicity type, primary treatment for RCC, time to local recurrence (months), tumor size at recurrence (mm), eGFR at salvage treatment (mL/min/1.73 m^2^), type of salvage treatment. The following perioperative data were recorded when available: clamping type, ischemia time, operative time, estimated blood loss (EBL), intraoperative complication (types and rates), length of stay (days), overall postoperative complication (defined according to Clavien–Dindo classification and rates). The following oncological and functional outcomes were collected: pathological stage, histology (types), positive margins, follow-up duration (months), recurrence, mortality, eGFR at last follow-up (mL/min/1.73 m^2^), pathological stage of recurrence, histology of recurrence, positive margins of recurrence.

### 2.4. Statistical Analysis

The meta-analysis was performed using ProMeta 3 software when there were two or more studies reporting the same outcome under the same definition. To calculate the pooled effect, a random effect model was applied. The effect size (ES) was reported with its 95% confidence interval (CI). Heterogeneity among studies was evaluated using the I^2^ statistics. A *p* < 0.05 was considered statistically significant. Egger’s linear regression test and Begg and Mazumdar’s rank correlation test were also used to evaluate the publication bias of studies included in the meta-analysis. Data not eligible for quantitative analysis were evaluated in a narrative fashion. The quality of the included studies was assessed using the Jadad Score or the Methodological Index for Non-Randomized Studies (MINORS) for randomized and non-randomized studies, respectively [18]. The JBI tool was adopted to evaluate the quality of case reports and case series [19]. Ethical approval and patient consent were not required for the present study.

## 3. Results

The search strategy revealed a total of 918 results. The screening of the titles and abstracts determined 398 papers eligible for inclusion. A further assessment of eligibility, based on the study of the full-text papers, led to the exclusion of 387 papers. Finally, 11 studies (4 case reports and 6 retrospective studies) involving 331 patients were included in the final analysis (Figure 1).

Of these, three studies involved only patients with a RCC recurrence after radiofrequency, cryoablation therapies, and stereotactic radiotherapy. Eight papers involved patients with a RCC recurrence after conservative surgical treatment.

### 3.1. Patient Demographics and Tumor Characteristics at Salvage Treatment

Study characteristics, patients’ demographics, and tumors’ features at salvage therapy are summarized in Table 1 and Table 2. The median age ranged from 44 to 75 years. Overall, 238 (71.9%) were males. Primary treatments for RCC included 267 (80.6%) PNs, 40 (12.0%) RFAs, 23 (6.9%) Cas, and 1 (0.3%) stereotactic radiotherapy. Specifically, among PNs, an open approach was mentioned in one retrospective study as well as in two case reports, while one paper mentioned a robot-assisted PN. No specific data on the approach used were presented for the remaining seven papers. Tumor stage was mentioned in five papers. Histology was mentioned in six papers. Additionally, only three papers mentioned the positive surgical margins (PSMs) for a total of 13 (3.9%) cases.

Overall, the median time of local recurrence for RCC patients treated with PNs ranged from 25 to 99 months; it was estimated at 13 months for RFA and CA treatments and at 6 months for stereotactic radiotherapy. The median recurrence size varied from 2 to 3.6 cm. The eGFR value before salvage therapy was discussed in seven papers, with a median value that ranged from 44.0 to 95.3 mL/min. Within the sample, 163 patients underwent salvage PN, 143 patients underwent RN, 22 patients underwent RFA, and 4 patients underwent CA at RCC local recurrence.

**Table 1 diagnostics-15-00838-t001:** Study and patient characteristics.

Study	Design	Year	Sample Size	Age, Years Median (IQR)	Gender (M/F)	Primary Treatment Type (*n*)	Pathological Stage Stage, *n* (%)	Positive Margins, *n* (%)
Antonelli [3]	R	2017	18	61.2 (13.1)	12/6	PN (18)	pT1a, 14 (77.7) pT1b, 1 (5.5) pT2a, 1 (5.5) pT3a, 2 (11.1)	12 (66.6)
Autorino [2]	R	2013	9	69 (61.5–71)	3/6	PN (9)	T1a 7 (78%) T3a 1 (11%) n.a. 1 (11%)	n.a
Watson [6]	R	2016	124	47.0 (28–75)	21/5	RPN (26)	n.a.	n.a.
Liu [7]	R	2010	25	51 (27–70)	13/12	OPN (25)	n.a.	n.a.
Gorin [10]	CR	2012	1	70	-/1	SR	-	-
O’Connor [9]	CR	2020	1	50	1/-	OPN	n.a.	n.a.
Perri [4]	CR	2021	1	75	-/1	OPN	pT1b, 2	0 (0)
Johnson [8]	R	2008	47	44 (20–70)	33/18	PN (47)	n.a.	n.a.
Nguyen [11]	R	2007	36	68 (40–83) 70 (62–84)	27/9	RFA (22) CA (14)	pT1a 6 pT3a 3 pT3b 2	n.a.
Jimenez [13]	R	2015	27	64 (57–71)	18/9	RFA (18) CA (9)	-	n.a.
Huang [15]	R	2023	140	n.a.	110/30	PN (140)	T1a, 97 (69%) T1b, 34 (24%) T2a, 3 (2%) T2b, 1 (0.7%) T3a, 5 (3.5%)	1 (0.7%)

The descriptive statistic has been reported as the median (IQR). When these values were unavailable, the mean and SD were reported. Abbreviations: CA, cryoablation; CR, case report; IQR, interquartile range; n.a., not available; OPN, open partial nephrectomy; PN, partial nephrectomy; SR, stereotactic radiotherapy; R, retrospective study; RFA, radiofrequency ablation.

**Table 2 diagnostics-15-00838-t002:** Recurrence characteristics.

Study	Recurrence, *n* (%)	Time to Local Recurrence, mo Median (IQR)	Local Recurrence Size, cm Median (IQR)	eGFR at Local Recurrence (mL/min) Median (IQR)	Salvage Treatment Type (*n*)	Follow-Up Duration, mo Median (IQR)
Antonelli [3]	18 (100)	44 (42)	3.48 (n.a.)	n.a.	RN (18)	**43.5** (n.a.)
Autorino [2]	9 (100)	39.4 (n.a.)	2.3 (1.2–3.6)	70.5 (n.a.)	RAPN (9)	8.3 (13)
Watson [6]	26 (20.9)	n.a.	n.a.	82.7 (34.92–103.03)	RAPN (26)	**3** (n.a.)
Liu [7]	25 (100)	99 (13–200)	3.5 (1.4–9)	53 (32–74)	OPN (25)	50 (3–196)
Gorin [10]	1	6	2.2	69	RAPN	n.a.
O’Connor [9]	1	n.a.	3.4	44	RAPN	n.a.
Perri [4]	1	48	2 and 2.2	69.0	OPN RFA	**18**
Johnson [8]	47 (100)	50 (n.a.)	3.5 (0.9–8.0)	95.3	PN (47)	**56** (n.a.)
Nguyen [11]	36 (100)	n.a.	3 (1.2–6.0) 2.9 (0.9–5.8)	n.a.	ORN (3) OPN (4) LRN (4) RFA (22) CA (3)	n.a.
Jimenez [13]	27 (100)	13 (8.5–24.5)	3.6 (2.2–4.6)	n.a.	OPN (15) ORN (5) LRN (7)	14.5 (n.a.)
Huang [15]	140 (100)	36 (n.a.) 25 (n.a.)	n.a.	n.a.	OPN (15) LPN (17) RAPN (2) ORN (53) LRN (52) RARN (1)	n.a.

The descriptive statistic has been reported as the median (IQR). When these values were unavailable, the mean and SD were reported. The bold numbers mean that the follow-up duration was calculated from the recurrence to the latest follow-up. Abbreviations: CA, cryoablation; eGFR, estimated glomerular filtration rate; IQR, interquartile range; LPN, laparoscopic partial nephrectomy; LRN, laparoscopic radical nephrectomy; n.a., not available; OPN, open partial nephrectomy; ORN, open radical nephrectomy; PN, partial nephrectomy; RFA, radiofrequency ablation; RN, radical nephrectomy; RAPN, robot-assisted partial nephrectomy; RARN, robot-assisted radical nephrectomy.

### 3.2. Perioperative Data

The perioperative data of salvage treatments performed are summarized in Table 3. Overall, the clamping type during PN was evaluated in five studies. Across these, 71 PNs adopted hilar clamping, 17 procedures were clamp-less, and in four procedures, an en block approach with Satinsky was used. Data concerning the clamping type on the remaining PNs were not available in six papers. When available, the median ischemia time ranged from 17.5 to 46 min with a median operating time that ranged from 153 to 510 min. The median EBL ranged from 225 to 2400 mL. Only six studies measured the median LOS, which ranged from 3 to 10 days. Overall, seven studies investigated the intraoperative complications, and nine and ten studies, respectively, investigated the overall postoperative complications and Clavien–Dindo ≥3 postoperative complications. Specifically, regarding the RCC patients who underwent salvage treatment, 53 (16.0%) subjects had intraoperative complications, while 103 (31.1%) patients had postoperative complications. According to Clavien–Dindo classification, 33 (9.9%) patients harbored grade ≥3 complications.

Additionally, oncological outcomes (PSM) were mentioned in seven papers. Specifically, six (1.8%) RCC patients exhibited PSM at salvage treatment. The median follow-up time post-recurrence ranged from 8.3 to 50 months. Overall, the median eGFR value at last follow-up after salvage therapy ranged from 35 to 81.6 mL/min/1.73 m^2^. The overall survival rate for RCC patients who underwent salvage therapy was 84.8%.

**Table 3 diagnostics-15-00838-t003:** Perioperative characteristics at salvage treatment.

Study	Recurrence *N*	Salvage Treatment Type (*n*)	Operative Time, min Median, (IQR)	EBL, mL Median, (IQR)	Length of Hospital Stay, Days Median, (IQR)	eGFR at Last Follow-Up (mL/min) Median (IQR)	Intraoperative Complications *n*, %	Overall Postoperative Complications *n*, (%)	Postoperative COMPLICATIONS According to CD ≥ 3 *N*, (%)	Pathological Stage of Local Recurrence *n*, (%)	Positive Margins of Recurrence, *n* (%)	Overall Survival, *n* (%)
Antonelli [3]	18	RN	n.a.	n.a.	n.a.	n.a.	0	1 (5.5)	1 (5.5)	pT1a, 6 (33.3) pT1b, 2 (11.1) pT3a, 7 (38.8) pT3b, 2 (11.1) pT4, 1 (5.5)	0	8 (44.4)
Autorino [2]	9	RAPN (2)	153 (105–236)	150 (75–275)	3 (2–3.5)	63.5 (n.a.)	n.a.	2 (22.2)	0	pT1a 7 (77.7) pT3a 1 (11.1) NA 1 (11.1)	n.a.	0
Watson [6]	26	RAPN (26)	359.5 (180–652)	900.0 (200–8500)	10.0 (5–19)	81.6 (39–120)	n.a.	15 (57.7)	3 (11.5)	n.a.	n.a.	n.a.
Liu [7]	25	OPN	510 (248–660)	2400 (800–14,000)	n.a.	49 (21–77)	20 (n.a.)	30 (n.a.)	16 (64)	n.a.	n.a.	24 (96)
Gorin [10]	1	RAPN	n.a.	100	5	57	0	0	0	pT1a	0	1
O’Connor [9]	1	RAPN	n.a.	100	3	n.a.	0	0	0	pT1a	0	1
Perri [4]	1	OPN	n.a.	n.a.	n.a.	35	0	0	0	n.a.	0	1
Johnson [8]	47	OPN	450 (240–840)	1800 (50–21,500)	n.a.	84.6	18 (38.2)	22 (46.8)	6 (12.7)	n.a.	n.a.	46 (97.9)
Nguyen [11]	36	ORN (3) OPN (4) LRN (4) RFA (22) CA (4)	214 (180–247) 284 (182–545)	150 (100–200) 710 (200–180)	n.a.	n.a.	6 (16.6)	2 (5.5)	n.a.	pT1a 6 (16%) pT3a 1 (2.7%) pT3b 1 (2.7%) pT3bN2 1 (2.7%) n.a. 27 (75%)	1 (2.7%)	36 (100)
Jimenez A [13]	27	OPN (15)	n.a.	353 (n.a.)	6 (n.a.)	n.a.	n.a.	6 (40)	4 (26.6)	pT1a 13 (50%) pT1b 5 (19%) pT3a 8 (31%)	0	15 (100)
Jimenez B [13]		RN (12)	n.a.	3157 (n.a.)	5 (n.a.)	n.a.	n.a.	4 (33.3)	2 (16.6)		0	12 (100)
Huang A [15]	140	PN (34)	150 (n.a.)	150 (n.a.)	n.a.	71.4 (n.a.)	1 (2.9)	5 (14.7)	0	pT1a, 78 (55.7) pT1b, 24 (17) pT2a, 3 (2) pT2b, 2 (1.4) pT3a, 33 (23.5)	1 (2.9)	34 (100)
Huang B [15]		RN (106)	158 (n.a.)	135 (n.a.)	n.a.	57.0	8 (7.5)	16 (15)	1 (0.9)		4 (3.7)	106 (100)

The descriptive statistic has been reported as the median (IQR). When these values were unavailable, the mean and SD were reported. Abbreviations: CA, cryoablation; CD, Clavien–Dindo classification; EBL, estimated blood loss; eGFR, estimated glomerular filtration rate; IQR, interquartile range; LRN, laparoscopic radical nephrectomy; n.a., not available; OPN, open partial nephrectomy; ORN, open radical nephrectomy; PN, partial nephrectomy; RFA, radiofrequency ablation; RN, radical nephrectomy; RAPN, robot-assisted partial nephrectomy.

### 3.3. Pooled Analysis for Overall Postoperative Complications

Six studies distinguished overall postoperative complication rates between PN and RN at salvage treatment (Figure 2). The overall pooled analysis showed an ES for overall postoperative complications of 0.28 (95% CI: 0.16–0.44; *p* < 0.001). When only PNs were considered (I^2^ = 69.35%, *p* = 0.011), the pooled ES for overall postoperative complications was 0.37 (95% CI: 0.22–0.54; *p* = 0.13). When only RNs were considered (I^2^ = 47.86%, *p* = 0.14), the pooled ES for overall postoperative complications was 0.17 (95% CI: 0.08–0.33; *p* = 0.001). No statistically significant differences emerged when the ESs were compared between PN and RN (*p* = 0.08). A bias evaluation is reported in Appendix A.

**Figure 2 diagnostics-15-00838-f002:**
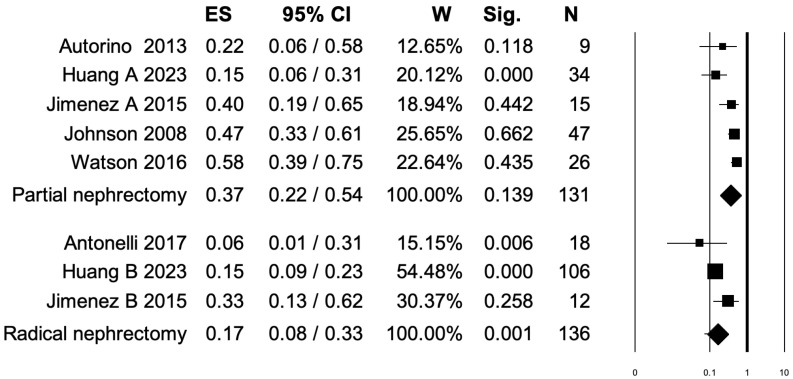
Forest plot showing the ES for overall postoperative complications [2,3,6,8,13,15]. ES: effect size; CI, confidence interval (overall I^2^ = 79.01%, *p* < 0.001).

### 3.4. Pooled Analysis for Clavien–Dindo Grade ≥ III Postoperative Complications

Seven studies distinguished Clavien–Dindo grade ≥ III postoperative complications rates between PN and RN at salvage treatment (Figure 3). The overall pooled analysis showed an ES for Clavien–Dindo grade ≥ III postoperative complications of 0.11 (95% CI: 0.04–0.28; *p* < 0.001). When only PNs were considered (I^2^ = 82.45%, *p* < 0.001), the pooled ES for Clavien–Dindo grade ≥ III postoperative complications was 0.17 (95% CI: 0.05–0.41; *p* = 0.012). When only RNs were considered (I^2^ = 65.3%, *p* = 0.056), the pooled ES for Clavien–Dindo grade ≥ III postoperative complications was 0.05 (95% CI: 0.01–0.24; *p* = 0.001). No statistically significant differences emerged when the ESs were compared between PN and RN (*p* = 0.2). A bias evaluation is reported in Appendix A.

**Figure 3 diagnostics-15-00838-f003:**
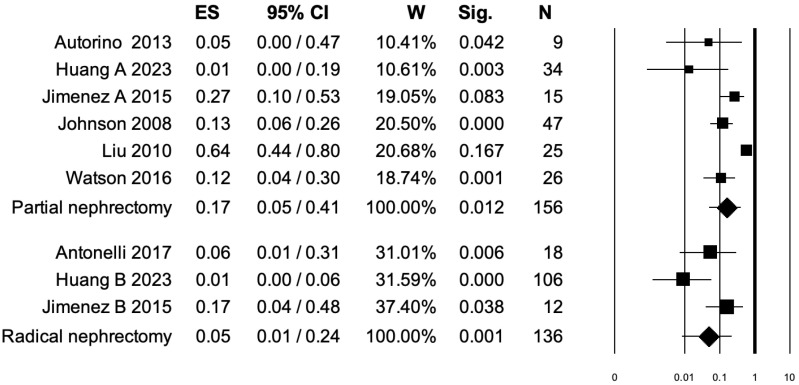
Forest plot showing the ES for Clavien–Dindo ≥ III postoperative complications [2,3,6,7,8,13,15]. ES: effect size; CI, confidence interval (overall I^2^ = 81.38%, *p* < 0.001).

### 3.5. Pooled Analysis for Intraoperative Complications

Four studies distinguished intraoperative complications rates between PN and RN at salvage treatment (Figure 4). The overall pooled analysis showed an ES for intraoperative complications of 0.18 (95% CI: 0.04–0.54; *p* = 0.075). When only PNs were considered (I^2^ = 90.68%, *p* < 0.001), the pooled ES for intraoperative complications was 0.34 (95% CI: 0.07–0.79; *p* = 0.51). When only RNs were considered (I^2^ = 0.0%, *p* = 0.45), the pooled ES for intraoperative complications was 0.07 (95% CI: 0.04–0.13; *p* < 0.001). No statistically significant differences emerged when the ESs were compared between PN and RN (*p* = 0.07). A bias evaluation is reported in Appendix A.

**Figure 4 diagnostics-15-00838-f004:**
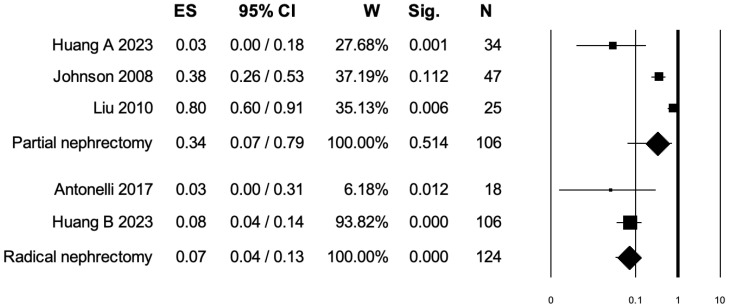
Forest plot showing the ES for intraoperative complications [3,7,8,15]. ES: effect size; CI, confidence interval (overall I^2^ = 92.30%, *p* < 0.001).

## 4. Discussion

Salvage treatment for local recurrence in RCC patients is challenging [3,6,11,12]. Specifically, either PN or RN and non-surgical ablative techniques, namely RFA or CA, are feasible techniques in these patients [7,13,14,15,16]. The urologist’s decision making relies on adverse RCC prognostic features and the patients’ comorbidities and life expectancy [7,13,17]. Unfortunately, the level of evidence for salvage treatment in locally recurrent RCC is still low and the grade of recommendation is weak [1]. As a result, the current study aimed to summarize available evidence about the oncological, surgical, and functional outcomes of salvage therapy after previous treatments for RCC.

First, we retrieved a total of 11 studies, including 331 RC patients who underwent salvage treatment. Specifically, the condition can either occur after PN (from 1% to 10%) or after kidney-sparing treatments, namely RFA or CA (from 5% to 8%) [2,3]. However, such scant data highlighted the need to summarize the evidence in order to provide urologists a reliable source for clinical decision making as well as patients informed consent. Undoubtedly, the RCC patients treated within the studies were mainly males (71.9%), with a median age that ranged from 44 to 75 years. Thus, the current systematic review may not allow us to discuss the feasibility of salvage treatment within an elder population with RCC. Nonetheless, the studies were mainly from Europe, not allowing for a specific evaluation of American or Asian populations.

Second, we focused on the oncological outcomes after salvage treatment for RCC. Interestingly, we found that the median time of local recurrence for RCC patients treated with PNs ranged from 25 to 99 months, followed by 13 months for RFA and CA treatments and 6 months for stereotactic radiotherapy. Moreover, the median recurrence size was mainly represented by pT1 tumors with a median diameter from 2 to 3.6 cm. These numbers enlightened how the local recurrence after primary treatment for RCC may occur regardless of the type of treatment performed. Specifically, the surgery may exhibit a longer recurrence-free survival (RFS) interval. According to previous reports [18,19], the local recurrence may be mainly attributable to PSM or tumor characteristics. Di Maida et al. observed 26 RCC patients treated with NSS techniques that experienced recurrence within a single center study from 2007 to 2013 [20]. Of those, 14 (53.8%) cases developed a recurrence at the level of the previous tumor resection bed [18]. Conversely, Abu Ghanem et al., relying on a multicenter study involving 3331 RCC patients treated with either RN or PN from 15 centers over a period of six years (2006–2011), found that the RFS was led by histological RCC subtypes [21]. Specifically, RFS at 5 years ranged from 78% to 91%, in, respectively, clear-cell RCC and chromophobe RCC patients [19]. Within our report, only three papers mentioned the PSM rate, which accounts for 1.8% of the overall RCC recurrent patients (*N* = 331). These low rates may support a procedural as well as oncological safety profile of salvage treatment for RCC. Unfortunately, direct comparisons as well as pooled analyses predicting PSM cannot be carried out due to the scant data recorded. Furthermore, Marconi et al. studied 505 patients with RCC recurrence after curative treatment within the RECUR multicenter database over a period of six years (2006–2011) [22]. Of those recurrent patients, 176 (34.8%) had a resectable disease [22]. Specifically, a resectable disease was defined as “solitary metastases or oligometastases or renal fossa or renal recurrence after PN/RN” [22]. Of those 176 patients, only 97 received local treatment, which showed a protective effect on overall mortality (odds ratios from 0.26 to 0.36) [22]. Interestingly, 79 patients (6%) were instead managed with palliative intent [22]. These data are crucial to identify patients who really will benefit from the salvage setting. Despite a proper surgical case selection based on patients’ characteristics (such as age, renal functional status, and comorbidities) as well as tumor characteristics (size and location), there are underlying characteristics that may play a crucial role in recurrence that we cannot account for. However, from our analysis, it emerged that the overall survival of salvage-treated RCC patients, regardless of the salvage treatment, was 84.8%. These results should be interpreted carefully due to the lack of follow-up data in four papers, which might underpower the analysis. Taken together, the safety profile of salvage treatment options may justify considering such procedures, namely surgery options as well as nephron-sparing treatment options, in a salvage setting. This concept should be stressed in clinical decision making as well as when obtaining patient consent. Nonetheless, as we did in the current study, it is essential to quantify the salvage treatment-associated morbidity.

Third, we analyzed the intraoperative complication rates among the studies collected. Overall, 53 (16%) RCC patients experienced intraoperative complications. Among these, blood transfusions, vascular injury, and pleural injury were very commonly experienced [13,14]. Liu et al. reported a higher rate of intraoperative blood transfusion (76%) within 25 salvage PN treated patients, followed by a vascular injury rate of 25% [7]. Conversely, Johnson et al. reported a rate of 64.7% for intraoperative blood transfusions within 47 RCC patients treated with salvage PN, followed by a rate of 19.1% for pleural injuries [8].

Fourth, we also addressed the overall postoperative complications and Clavien–Dindo grade ≥ III postoperative complication rates of RCC recurrent patients who underwent treatment. Specifically, postoperative complications occurred in 103 (31.1%) RCC patients. Of those, blood transfusions and hemodialysis were the most discussed [11,13,14]. Specifically, the postoperative blood transfusion rate was 20% for Liu et al. [7] while almost 10% for Johnson et al. [8]. These rates are lower than the intraoperative blood transfusion rates. However, important parameters such as hemoglobin drop and preoperative anemia status are missing. Additionally, the rates of patients who received both intra- and postoperative blood transfusion are also missing. Nonetheless, blood loss was not reported for all the studies collected. Such important information may introduce biases in the evaluation of blood transfusions rate as well as in further data interpretations. Furthermore, hemodialysis rates were reported for 3.9% to 12% of cases [13,14]. However, the eGFR drop is also missing in those reports, rendering further subgroup analysis impossible. Interestingly, of all postoperative complications, only 33 (9.9%) RCC patients harbored Clavien–Dindo grade ≥ III postoperative complication rates. Due to the presence of those rates of complications at PN more than RN, it is imperative to perform a pooled subgroup analysis addressing each outcome within PN and RN subgroups.

Fifth, three independent pooled analyses were performed to assess the effect of salvage treatment (either PN or RN) on overall postoperative complications, Clavien–Dindo grade ≥ III postoperative complications, and intraoperative complications. We observed a protective, albeit weak, association between RN and overall postoperative complications (ES: 0.17, *p* < 0.001). Within RN subgroups, the I2 was also low (47.83%), strengthening the clinical meaningfulness of our analysis. No statistically significant association was found between the PN subgroup and overall postoperative complications. Conversely, when the endpoint of interest was the Clavien–Dindo grade ≥ III postoperative complications, both PN (ES: 0.17, *p* = 0.012) and RN (ES: 0.05, *p* = 0.001) exerted a protective, albeit weak, association. Finally, when the last outcome, namely intraoperative complications, was analyzed, we observed an ES of 0.07 only for RN (*p* < 0.001). These results may be interpreted as a surrogate of feasibility of salvage PN as well as RN in RCC patients. Indeed, despite the ESs being weak, RN invariably exhibited an acceptable profile of safety for overall postoperative complications as well as Clavien–Dindo grade ≥ III and intraoperative complications. Additionally, PN also exhibited an acceptable profile of safety for Clavien–Dindo grade ≥III postoperative complications.

Taken together, the safety profile of salvage treatment options may justify considering such procedures, namely PN or RN as well as nephron-sparing treatments, in a salvage setting. First, a proper surgical case selection based on patients’ characteristics (such as age, renal functional status, and comorbidities) as well as tumor characteristics (size and location) is imperative. Second, there are underlying characteristics beyond PSM and histological variants that may play a crucial role in recurrence that we cannot account for. Third, the overwhelming majority of RCC patients after salvage treatment survived (OS rate = 84.8%). Nonetheless, the morbidity profile was tolerable with a rate of complications that ranged from 16% for intraoperative to 31.1% for overall postoperative complication rates, with less than 10% represented by major complications (Clavien–Dindo grade ≥ III). These valuable aspects should be stressed in clinical decision making as well as for purposes involving patient consent. However, further larger studies that will overcome the limitations enlightened in the current review are needed to improve the evidence on salvage RCC treatment options.

Despite the novelty of the current review, the potential limits of the available literature must be acknowledged: the available studies are few, small-sized, retrospective, not controlled, and with short and/or missing follow-up. Moreover, they encompass a wide time range with significant heterogeneity in reporting intra- and postoperative complication rates. Nonetheless, they lack specifical information on surgical approached performed (robotic vs. open) and biochemical data (such as hemoglobin values and kinetics) that may affect the final results. Undoubtedly, the paucity of studies as well as perioperative data also hindered subgroup analyses that might have added more information to the current study. However, these limitations would likely be shared with future systematic analyses, due to the rarity of the condition, which could be found in more in-depth and prospective-maintained database. Finally, further, wider series are needed to better investigate the oncological as well as the functional profile of salvage treatment for recurrent RCC.

## 5. Conclusions

The profile of oncological safety as well as either the morbidity or mortality of salvage treatment options for RCC may justify considering such procedures in a salvage setting. This concept should be stressed in clinical decision making as well as for purposes involving patient consent. However, further larger studies that will overcome the limitations enlightened in the current review are needed to improve the evidence on salvage RCC treatment options.

## Figures and Tables

**Figure 1 diagnostics-15-00838-f001:**
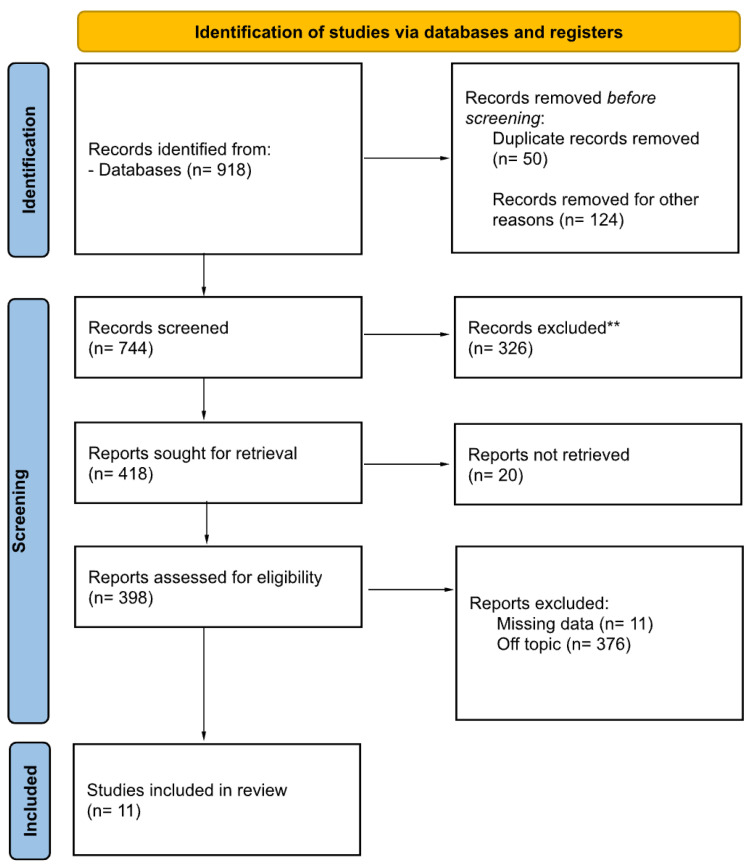
PRISMA 2020 flow diagram for new systematic reviews, which included searches of databases, registers, and other sources. ** They were excluded after reading the abstract.

## Data Availability

The datasets generated during and/or analyzed during the current study are available from the corresponding author on reasonable request.

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
