# Peer review of "Outcomes of Salvage Treatment After Primary Treatment for Renal Cell Cancer: A Systematic Review"

_diagnostics, 2025, doi:10.3390/diagnostics15070838_

Round 1
Reviewer 1 Report
Comments and Suggestions for Authors
The problem of recurrences after primary tratment for RCC is a real issue and deserve to be studied.This systematic review is well written,and understandable.I have the following comments:
-In the Introduction part,the vast majority of statements were taken from EAU Guidelines.But,in my opinion,the authors must cite the studies that EAU Guidelines had taken into consideration.
-in Method part,it is stated that two authors have read 318 articles,full text,which is very impresive.Probably,did you mean that you have screened the abstracts?
-The authors have found 4 case reports and 6 retrospective studies,and no prospective studies,which can be a drawback in taking the conclusions.Also,from these,only 4 studies are published after 2021,the rest are older,even 15 years ago,which can make questionable the novelty of the results.
-In table 2,I see only 3 case reports presented,not 4.
-Table 1 does not appear in your study
-row 336-337 "RCC patients survived after salvage treatment".How did you appreciated the survival rate,what was the followup period?Please specify in text.
-row 354-355 "considering such procedures". Please replace with RN and PN.
Author Response
We thank the reviewers and the editorial board for the opportunity to consider our work. Moreover, we are grateful for the constructive comments and the effort of each reviewer and the editorial team to evaluate our work within short time. Thank you for your time. We addressed the comments step by step.
#REVIEWER 1
The problem of recurrences after primary tratment for RCC is a real issue and deserve to be studied.This systematic review is well written,and understandable.I have the following comments:
Q1. In the Introduction part,the vast majority of statements were taken from EAU Guidelines.But,in my opinion,the authors must cite the studies that EAU Guidelines had taken into consideration.
A1. We thank the reviewer for the positive comments. We do agree with the reviewer and we implemented the citations in the Introduction part.
Q2. In Method part,it is stated that two authors have read 318 articles,full text,which is very impresive.Probably,did you mean that you have screened the abstracts?
A2. Thank you for your comment. We meant “abstract” and in consequence we used the neutral term “record”.
Q3. The authors have found 4 case reports and 6 retrospective studies,and no prospective studies,which can be a drawback in taking the conclusions.Also,from these,only 4 studies are published after 2021,the rest are older,even 15 years ago,which can make questionable the novelty of the results.
A3. We thank the reviewer for the pertinent comment. We do agree with the oldness of some studies. However, the results that we are furnishing with the current manuscript were not discussed before. In such context, the systematic review, despite was based on retrospective cohort studies, achieved a pivotal point of discussing on the role of salvage treatment in RCC patients. Unfortunately, due to the rarity of the condition itself, no prospective study could be maintained to enlighten which is the best treatment option for such patients. As a result, the current analysis represents the first comprehensive description of the salvage treatment within RCC patients.
Q4-5-6-7. In table 2,I see only 3 case reports presented,not 4.
-Table 1 does not appear in your study
-row 336-337 "RCC patients survived after salvage treatment".How did you appreciated the survival rate,what was the followup period?Please specify in text.
-row 354-355 "considering such procedures". Please replace with RN and PN.
A4-7. We thank the reviewer for the comments, and we apologized for the errors. We corrected all the mistakes pointed out within the whole manuscript. Table 1 was added and simplified in two tables to display all the important variables. According the follow-up period, all the studied pointed out that follow-up period was from salvage treatment to beyond. For the available studies, it has been recorded in Table 1. Otherwise, in the main text, we re-write the sentence which now reads as follow: “However, from our analysis, it emerged that the overall survival of salvage treated RCC patients regardless the salvage treatment was 84.8%. Those results should be interpreted carefully due to the lacking in follow-up data in four papers, that might underpowered the analysis.”
Reviewer 2 Report
Comments and Suggestions for Authors
This article conducted a systematic review of the outcomes of salvage therapy after initial treatment for renal cell carcinoma (RCC), summarizing the evidence on surgical, oncologic, and functional outcomes. The structure of the article was clear, the research methods were in line with PRISMA guidelines, and the data extraction and analysis process were transparent. However, the article has some limitations that need to be improved.
1.The number of included studies was small (11), and most of them were retrospective studies. There was no randomized controlled trial (RCT) or prospective study, and the level of evidence was low. If possible,prospective studies and RCTS are recommended to be included to improve the reliability of the evidence.
2.There was high heterogeneity (high I² values) among the studies, especially in the analysis of some complications, which may affect the reliability of the results.I recommend you conduct further subgroup analyses or sensitivity analyses of the results with higher heterogeneity to identify the source of heterogeneity.
3.Some key data were missing, such as the type of surgery (open vs. robot-assisted), preoperative anemia status, and decrease in hemoglobin, which may have affected the interpretation of the results.If certain key data are not reported in the original literature, you can attempt to contact the authors of the original literature to obtain missing data.Or you can clearly state in the discussion section which data are missing and discuss the impact these missing data may have on the results.
4.The resulting graphs are too few, too simple, and too single to present all the valid data.It is recommended that the authors add more graphs to show other key outcomes such as oncologic outcomes (e.g., relapse-free survival, overall survival) and functional outcomes (e.g., long-term changes in renal function). These graphs can help readers understand the data more intuitively. In addition to forest plots, authors may consider using other types of graphs, such as Kaplan-Meier curves (to show survival rates), boxplots (to show changes in renal function), etc., to present the data more comprehensively.
Author Response
We thank the reviewer for the positive comments.
Q1.The number of included studies was small (11), and most of them were retrospective studies. There was no randomized controlled trial (RCT) or prospective study, and the level of evidence was low. If possible,prospective studies and RCTS are recommended to be included to improve the reliability of the evidence.
A1. We thank the reviewer for the intriguing comment. Unfortunately, due to the rarity of the condition itself, no prospective studies as well as RCTs could be maintained to enlighten which is the best treatment option for such patients. As a result, the current analysis represents the first comprehensive description of the salvage treatment within RCC patients.
Q2. There was high heterogeneity (high I² values) among the studies, especially in the analysis of some complications, which may affect the reliability of the results.I recommend you conduct further subgroup analyses or sensitivity analyses of the results with higher heterogeneity to identify the source of heterogeneity.
A2. We thank the reviewer for the pertinent comment. Unfortunately, the paucity of the studies does not allow us to proceed granularly. No further subgroup analysis could be performed. We acknowledged this evidence in limitations section.
Q3.Some key data were missing, such as the type of surgery (open vs. robot-assisted), preoperative anemia status, and decrease in hemoglobin, which may have affected the interpretation of the results.If certain key data are not reported in the original literature, you can attempt to contact the authors of the original literature to obtain missing data.Or you can clearly state in the discussion section which data are missing and discuss the impact these missing data may have on the results.
A3. We thank the reviewer for the comment. We pointed out those critical and correct opinion in the limitation section which now reads as follow: “Despite the novelty of the current review, the potential limits of the available literature must be acknowledged: the available studies are few, small-sized, retrospective, not controlled, and with short and/or missing follow-up. Moreover, they encompass a wide time range with significant heterogeneity in reporting intra- and postoperative complication rates. Nonetheless, they lack specifical information on surgical approached performed (robotic vs. open), biochemical data (such as hemoglobin values and kinetics) that may affect the final results. Undoubtedly, the paucity of studies as well as perioperative data also hindered subgroup analyses that might be added more information to the current study. However, those limitations would likely be shared with the future sys-tematic analyses, due to the rarity of the condition, that could be found in further and prospective-maintained database. Finally, further, wider se-ries, are needed to better investigate the oncological as well as the func-tional profile of salvage treatment for recurrent RCC.”.
Q4.The resulting graphs are too few, too simple, and too single to present all the valid data.It is recommended that the authors add more graphs to show other key outcomes such as oncologic outcomes (e.g., relapse-free survival, overall survival) and functional outcomes (e.g., long-term changes in renal function). These graphs can help readers understand the data more intuitively. In addition to forest plots, authors may consider using other types of graphs, such as Kaplan-Meier curves (to show survival rates), boxplots (to show changes in renal function), etc., to present the data more comprehensively.
A4. We thank the reviewer for the pertinent comment. Unfortunately, those graphs are not reproducible due to the lack in that information within the manuscript cited (relapse-free survival, overall survival). Furthermore, we do not report functional outcome such as changes in renal function due to the stronger importance of oncological outcomes in RCC patients underwent salvage treatment.
Round 2
Reviewer 2 Report
Comments and Suggestions for Authors
The author has made the changes as required, and the reasons for not being able to be modified are also stated where they cannot be modified. My final recommendation is to agree to publication.